# Characterization of hemodialysis fistulas experienced abrupt thrombosis and determination of a proper follow-up protocol: A retrospective cohort study and an interventionist's perspective

**Matt Chiung-Yu Chen**[1]*, **Mei-Jui Weng**[2], **Lee-Hua Chao**[1]

**1** Department of Interventional Radiology, Yuan's General Hospital, Kaohsiung, Taiwan, **2** Department of Radiology, Kaohsiung Veterans General Hospital, Kaohsiung, Taiwan

* jjychen@gmail.com

**Data Availability Statement:** All data have been deposited to Figshare (doi: 10.6084/m9.figshare.

## Abstract

Abrupt thrombosis is a form of thrombosis that occurs unexpectedly and without being preceded by hemodialysis fistula (AVF) dysfunction during dialysis. We found that AVFs with a history of abrupt thrombosis (abtAVF) appeared to have more episodes of thrombosis and required more frequent interventions than those without such history. Therefore, we sought to characterize the abtAVFs and examined our follow-up protocols to determine which one is optimal. We performed a retrospective cohort study using routinely collected data. The thrombosis rate, AVF loss rate, thrombosis-free primary patency, and secondary patency were calculated. Additionally, the restenosis rates of the AVFs under the follow-up protocol/ sub-protocols and the abtAVFs were determined. The thrombosis rate, procedure rate, AVF loss rate, thrombosis-free primary patency, and secondary patency of the abtAVFs were 0.237/pt-yr, 2.702/pt-yr, 0.027/pt-yr, 78.3%, and 96.0%, respectively. The restenosis rate for AVFs in the abtAVF group and the angiographic follow-up sub-protocol were similar. However, the abtAVF group had a significantly higher thrombosis rate and AVF loss rate than AVFs without a history of abrupt thrombosis (n-abtAVF). The lowest thrombosis rate was observed for n-abtAVFs, followed up periodically under the outpatient or angiographic sub-protocols. AVFs with a history of abrupt thrombosis had a high restenosis rate, and periodic angiographic follow-up with a mean interval of 3 months was presumed appropriate. For selected populations, such as salvage-challenging AVFs, periodic outpatient or angiographic follow-up was mandatory to extend their usable lives for hemodialysis.

## Introduction

In practice, we observed a subgroup of hemodialysis fistulas (AVFs) with a history of abrupt thrombosis (abtAVF), requiring more frequent interventions for AVF thrombosis than those

21996920), which can be accessed without restriction.

**Funding:** The author(s) received no specific funding for this work.

**Competing interests:** The authors have declared that no competing interests exist.

**Abbreviations:** abtAVF, AVF with a history of abrupt thrombosis; AVF, Arteriovenous fistula; n-abtAVF, AVF without a history of abrupt thrombosis; ROC, Receiver-operator curve; VA site, hemodialysis vascular access site.

without such history (n-abtAVF). Abrupt thrombosis is a form of thrombosis that occurs unexpectedly and without being preceded by AVF dysfunction during dialysis.

Based on the "dysfunction hypothesis" [1], surveillance depends on the concept that stenosis causes observable hemodialysis vascular access (VA) site dysfunction, and this dysfunction reliably precedes and accurately predicts thrombosis. In theory, all VA sites should present dysfunction before occurrence of thrombosis. However, we noticed that dysfunction hypothesis didn't apply to the abtAVFs, as no AVF dysfunction precedes thrombosis.

Clinically, a >50% stenosis can be innocent without incurring VA site dysfunction. Tessitore et al. [2] found that performing flow surveillance with subsequent preemptive treatment of subclinical stenosis in AVFs can reduce thrombosis rate and prolong their useful lives for dialysis. However, the recommendation to treat all subclinical stenoses is still controversial on its cost-effectiveness. To address this, adoption of a more stringent criteria to treat a subclinical >50% stenosis only when it poses a high risk of thrombosis might be a reasonable option.

The Spanish guidelines [3] proposed the concept of the thrombosis-high-risk stenosis (THRS), which was defined as a stenosis of >50% or with a residual luminal diameter of <2mm combined with additional criterion suggesting severe hemodynamic derangements. Asymptomatic or subclinical THRS was presumed as a probable cause of abrupt thrombosis of AVFs [4, 5]. For AVFs where stenosis did not cause AVF dysfunction, detection of THRS, and performance of preemptive treatment were required to prevent abrupt thrombosis. The 2000 Kidney Disease Outcomes Quality Initiative (K/DOQI) update on vascular access [6] mentioned that individual patients may have rapid recurrence of stenosis that requires repeated PTA, which implied the importance of restenosis rate of a stenosis in design of a surveillance protocol. We hypothesized that the abtAVFs had a fast restenosis rate and that our current follow-up protocol failed to alert us in time to perform preemptive treatment before the occurrence of thrombosis. Unlike hemodialysis staff, interventionists do not see patients once every 2 days. Therefore, patients must be called back periodically to observe the severity of stenosis and determine when it should be treated. The time interval for patients with an AVF to be called back following angioplasty should be based on the restenosis rate. Stenoses with faster restenosis rates require a shorter treatment interval, and more frequent interventions. We therefore conducted the present retrospective cohort study to characterize abtAVFs. Additionally, we examined our follow-up protocols and sought to determine which one is optimal for the abtAVFs.

## Materials and methods

### Definitions

**AVF with abrupt thrombosis (abtAVF):** A thrombotic AVF with at least a >50% stenosis identified during ultrasound or angiography. The thrombosis occurred unexpectedly and without a preceding dysfunction, such as inadequate inflow or high dynamic venous pressure during hemodialysis. **AVF without a history of abrupt thrombosis (n-abtAVF):** These AVFs could have been dysfunctional or thrombotic. The thrombosis usually occurred by deferred correction of the culprit stenosis for AVF dysfunction. **Positive quantitative physical examination indicators (qPE indicators)** can detect severe stenosis-related hemodynamic derangement in stenoses with minimal luminal stenosis area of <2 mm [1]. To save time, we did not perform a systemic physical examination (PE). Rather, we focused on the detection of qPE indicators, which included: 1) No pulse on palpation over the arterial cannulation site; 2) No detection of a pulse downstream of the stenosis in the inflow segment of an AVF (sPPL = 0); and 3) An outflow score of 1- or PE significant outflow stenosis (PESOS) for detecting outflow stenosis. **Thrombosis-high-risk-stenosis (THRS):** In the literature, THRS is a stenosis that is

both anatomically (>50% or minimal luminal diameter <2 mm) and hemodynamically (Qa<350, PSVR>2, dQa>25%) significant (2). THRS was considered hemodynamically significant in the present study when a positive qPE indicator was detected. **Index of patency function (IPF)**: IPF was the time interval between the index procedure at enrollment and the end of the study or AVF abandonment divided by the number of interventions to maintain the access circuit for hemodialysis. IPF can be considered the mean time between reinterventions (3). **Restenosis interval and restenosis rate**: Restenosis interval is the time from when a stenosis is fully dilated to when it is narrowed again to meet the action criteria for angioplasty. For a given period, the shorter the stenosis interval, the higher the restenosis rate. *The protocol* **rate** refers to the restenosis rate for AVFs under a given follow-up protocol. These are the on-demand, outpatient, and angiographic sub-protocols in the present study; *the target rate* refers to the restenosis rate for a given follow-up group of AVFs, such as the abtAVF and n-abtAVF (periodic) groups. The protocol rate should be greater than or equal to the target rate to allow time for the THRS to be dilated before AVF thrombosis occurs.

**Patient population and study design.** Between November 2020 and February 2021, patients referred to our institution for treatment of vascular access (VA) sites were enrolled in this study. All VA sites were treated and followed according to our routine protocols. After obtaining approval from our hospital's institutional review board committee (IRB number: 20220011B), the patients' electronic imaging and medical records were reviewed. Data for this study was collected prospectively and routinely recorded after each treatment was completed. To conduct this study, the collected data was retrieved and analyzed retrospectively. Inclusion criteria for patients enrolled in this study were as follows: 1) Mature AVF (>6 months old) that was superficial and visible at least around the arterial and venous cannulation segments; 2) Dysfunctional AVFs that had been referred to us because of inadequate inflow (Qb <250 ml/min), elevated dynamic intra-access pressure (>180–200 mmHg) during dialysis or prolonged needle-site bleeding following dialysis, difficult needling, or inadequate clearance; 3) AVFs without dysfunction during dialysis that had been referred to us for treatment because abnormal PE results suggested the presence of AVF stenosis, including abnormal thrill and pulse on palpation, or decreased/abnormal bruit on auscultation. The following VA sites were excluded: AVGs, hemodialysis catheters, immature fistulas, poor candidates for performing a PE to detect stenosis (calcification, deeply located), and AVFs that were treated only once in our angiographic suite. Additionally, thrombotic AVFs without a detectable >50% stenosis on angiography or sonography were also excluded.

**Clinical indicators for detection of stenoses and the action criteria for angioplasty.** In the present study, the clinical indicators described below were used for clinical monitoring (Table 1). The accuracy of qPE indicators for detecting <2-mm stenoses was recently verified [5].

**Follow-up protocol and sub-protocols for follow-up of AVFs after angioplasty.** Our follow-up protocol includes three sub-protocols for different situations that may be encountered during AVF follow-up.

1. **The on-demand (od) sub-protocol** (Fig 1A): This sub-protocol was for detecting stenoses by clinical monitoring. Patients followed up under this protocol returned for an outpatient AVF check only when positive clinical indicators such as AVF dysfunction were observed during dialysis, or there were abnormal findings on PE. If a culprit stenosis was detected on sonography, the stenosis was slated for angioplasty.

2. **The outpatient (opd) sub-protocol** (Fig 1B): This sub-protocol was for detecting stenoses for preemptive intervention by periodic outpatient ultrasound and PE check of AVFs. Patients under this protocol generally returned at an interval of 2–3 months. At the

**Table 1.  Diagnostic elements of the clinical indicators.**

|  | Inflow stenosis | Outflow stenosis | Needling site stenosis | Thrombosis and others |
|---|---|---|---|---|
| **qPE indicators** | 1) No pulse on compression over the arterial needling area; 2) No pulse detected downstream of the detected inflow stenosis (sPPL = 0). | 1) Bounding pulse with no thrill on finger compression, and upon arm elevation, there was a systolic-only thrill (1-) or no thrill (PESOS). | A localized jet thrill at or around the needling site. | Hard on palpation without thrill/pulse or bruit. |
| **Hemodialysis indicators (4)** | 1) Inability to achieve the target dialysis blood flow, tube shaking; 2) >20%–25% decrease of Qb. | 1) Prolonged bleeding; 2) Dynamic venous pressure >180–200 mmHg. | 1) Aspiration of clots; 2) New difficulty with cannulation when previously not a problem. | Kt/V <1.2 or URR <65. |

The action criteria for angioplasty were as follows: 1) A >50% stenosis with at least one positive hemodialysis indicator; 2) A >50% or <2mm stenosis with at least one positive qPE indicator but no positive hemodialysis indicator.

outpatient visit, if point-of-care ultrasound detected a THRS, angioplasty was slated whether AVF dysfunction occurred or not.

3. **The angiographic (ang) sub-protocol** (Fig 1C): This sub-protocol was for performing pre-emptive intervention by detecting and treating any >50% stenoses during the same session as the angiographic study. Patients under this protocol returned periodically for angiographic checks of AVF stenosis, generally at an interval of 2–4 months. If a >50% stenosis was detected on angiography, subsequent angioplasty was performed.

**Selection of follow-up sub-protocols (Fig 2).**  abtAVFs were eligible for periodic follow-up under either the opd or ang sub-protocols. For n-abtAVFs and IPF $\geq$ 4 months, patients could choose to be followed up under the od or bi-monthly opd sub-protocol. For n-abtAVFs with IPF< 4 months, we adopted the ang or monthly opd sub-protocol.

**Follow-up outcome measures for sub-protocols and the target groups.**  The following parameters were used to assess the follow-up outcomes of VA site: 1) Thrombosis rate and thrombosis-free primary patency, 2) AVF loss rate and secondary patency, and 3) The restenosis rate for AVFs under each sub-protocol (the protocol rates) and the target groups (the target rates).

**Identification and characterization of the abtAVFs.**  We reviewed the electronic medical records and referral sheets of patients enrolled during the study period. For thrombotic AVFs followed up under the od sub-protocol, we routinely questioned the patients or their dialysis staff, and reviewed their referral sheets to determine if there were any dysfunctions during dialysis, episodes of low systemic blood pressure (<90 mmHg), or overzealous compression of the access site before the AVF thrombosis occurred. If none of these conditions were present, the AVF was considered an abtAVF. We routinely noted this in patients' electronic procedure reports, and following treatment, these abtAVFs were followed up periodically, according to either the opd or ang sub-protocol.

The outcome measures of abtAVFs were compared with those of other target groups, including the n-abtAVF, n-abtAVF(od), and n-abtAVF(periodic) groups. As to the restenosis rate, we did not report restenosis rates as did Razdan et al. [7] by calculating the % luminal reduction per day or month. However, it was possible to determine the relative restenosis rate, either faster or slower, between two given groups, according to the following rules:

1. For two groups with insignificantly different thrombosis rates and procedure rates, they had a similar restenosis rate.

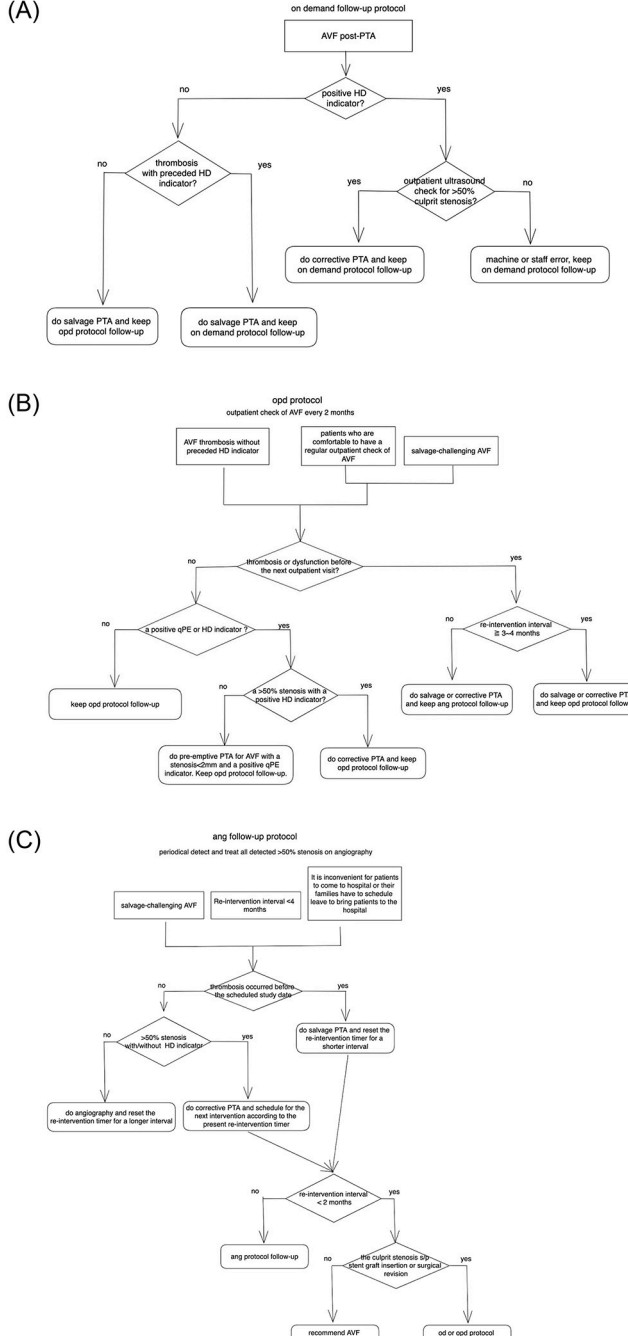

**Fig 1. The three follow-up sub-protocols.** (A) The on-demand (od) sub-protocol. (B) The outpatient (opd) sub-protocol. (C) The angiographic (ang) sub-protocol.

2. For two groups with insignificantly different thrombosis rates, the group requiring a higher procedure rate had a faster restenosis rate.

3. For two groups with insignificantly different procedure rates, the group that yielded a higher thrombosis rate had a faster restenosis rate.

**Fig 2. Flow chart for sub-protocol selection.**

4. For two groups with similar restenosis rates, the group with a higher procedure rate yielded a lower thrombosis rate.

## Statistical analysis

Thrombosis rate, AVF loss rate, and procedure rate were calculated and are presented as the number of events/patient-year (/pt-yr). Poisson's analysis was used to estimate differences in

rates. The thrombosis-free primary patency and secondary patency were calculated using survival analysis with the Kaplan–Meier method. Between-group comparisons of IPF were performed using the unpaired t-test. The receiver operating characteristic (ROC) analysis and the aforementioned analyses were performed using Prism Version 9.0 for Mac software (Graph-Pad Software, La Jolla, CA, USA).

## Results

A total of 505 VA sites in 505 patients were treated during the study period. Of them, 227 AVFs were enrolled. The study flow diagram is shown in Fig 3. Most of the excluded VA sites were AVGs, n = 149 (29.5%).

There were 93 male and 134 female with a mean age of 62.1±1.6 years, ranged from 24 to 87. As to the AVF types, there were 151 radiocephalic, 64 brachiocephalic, 9 brachiobasilic and 3 Gracz's AVFs. The patient demographic data were listed in Table 2.

### The follow-up outcomes of the sub-protocols

The follow-up outcome measures for the protocol (overall) and for each sub-protocol are shown in Fig 4. The follow-up outcomes for the protocol were as follows: for AVFs with a restenosis interval of ≥ 4 months followed under the od or opd sub-protocol, a procedure rate of 1.587–1.872/pt-yr may maintain the thrombosis rate at 0.072–0.086/pt-yr, whereas for AVFs with a restenosis interval of < 4 months followed under the ang sub-protocol, a procedure rate of 3.294/pt-yr may maintain the thrombosis rate at 0.060/pt-yr. The thrombosis rate among the three sub-protocols was not statistically different, indicating the performances of the three sub-protocols for prevention of AVF thrombosis were equivalent. The ang sub-protocol required the highest procedure rate and the od sub-protocol required the lowest. Moreover, the IPF of the ang sub-protocol was the shortest at 103.5 days, while that of the od sub-protocol was the longest at 234.4 days.

The thrombosis-free primary patency was similar among the three sub-groups (Fig 5A). Regarding AVFs in the ang sub-protocol, the secondary patency was significantly inferior to that in the od sub-protocol (Fig 5B).

**The characteristics of the abtAVFs.** A total of 33 (14.5%) abtAVFs were identified. The outcome measures of abtAVFs were compared with those of other target groups, including the n-abtAVF, n-abtAVF(od), and n-abtAVF(periodic) groups. The thrombosis rate, AVF loss rate, procedure rate, and IPF for abtAVF and the other target follow-up groups are shown in Figs 6A and 6B.

Among all target follow-up groups, the abtAVF group had the highest thrombosis rate (0.237/pt-yr) and AVF loss rate (0.027/pt-yr). The n-abtAVF(periodic) group had the lowest thrombosis rate (0.014/pt-yr). The relative protocol rates and target rates from highest to lowest (fast to slow) were as follows: abtAVF≒ang>opd>od = n-abtAVF(od)>overall≒n-abtAVF≒n-abtAVF(periodic).

The IPF of the abtAVF, n-abtAVF, and n-abtAVF(periodic) groups were 115.6 days, 237 days, and 149.8 days, respectively.

The thrombosis-free primary patency at 1, 4, and 8 years for each target follow-up group is shown in Table 3, and the survival plot is shown in Fig 7A. The abtAVF group had the lowest 1-year thrombosis-free primary patency (78.3%), and the n-abtAVF(periodic) group had the highest (100%).

The AVF loss rate and secondary patency of the abtAVF group and other target follow-up groups are shown in Table 4.

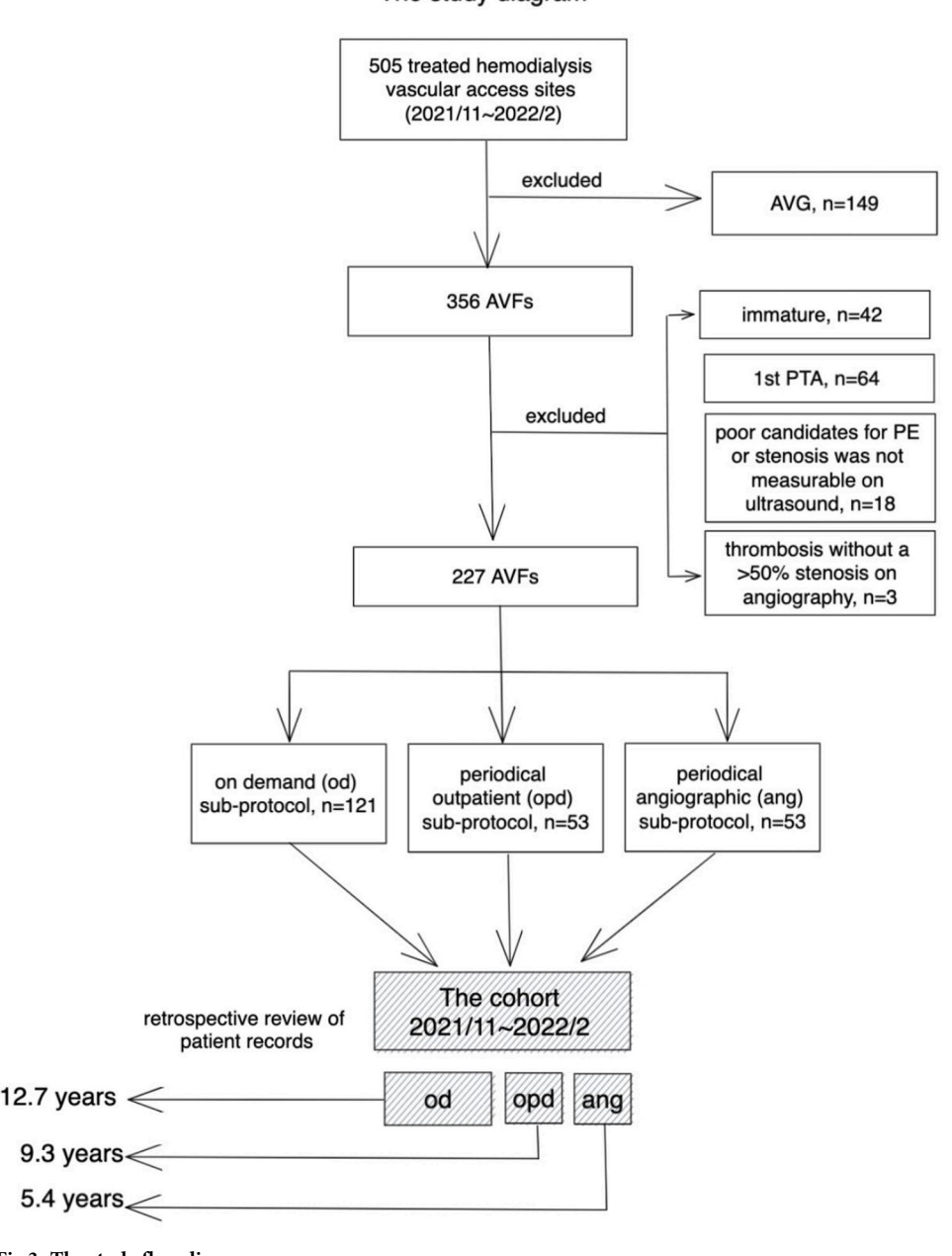

**Fig 3. The study flow diagram.**

The abtAVF group had a significantly higher AVF loss rate (0.027/pt-yr) than the other target follow-up groups (0.02~0.06 pt/yr). Regarding the secondary patency, no significant differences were observed among the target follow-up groups (Fig 7B).

A total of 74 episodes of thrombosis were noted in this study, and 68 salvage procedures were successfully performed (salvage success rate = 91.9%). A total of six AVFs were abandoned, four under the ang sub-protocol (66.7%), one under the opd sub-protocol, and one under the od sub-protocol. Additionally, two abandoned AVFs were noted in the abtAVF

**Table 2. Patient demographic data.**

|  | Age (years old) | Gender | AVF type |
|---|---|---|---|
| **od** | 62.0±2.2 | M(59); F(62) | RC(87), BC(28), BB(5), GZ(1) |
| **opd** | 60.6±3.7 | M(21), F(32) | RC(39), BC(13), GZ(1) |
| **ang** | 64.1±2.8 | M(13), F(40) | RC(25), BC(23), BB(4), GZ(1) |
| **abtAVF** | 65.0±2.1 | M(12), F(21) | RC(23), BC(9), BB(1) |
| **n-abtAVF** | 61.1±1.4 | M(22), F(51) | RC(41), BC(27), BB(3), GZ(2) |

M() = Male(patient count); F() = Female(patient count); RC = radiocephalic AVF; BC = brachiocephalic AVF; BB = brachiobasilic AVF; GZ = Gracz's AVF.

group (one thrombotic mega-fistula and one rapid restenosis) and four in the n-abtAVF group. The causes of AVF loss were as follows:

1. Rapid recurrence of stenosis (n = 2, ang sub-protocol). For one patient, the on-duty surgeon preferred to create a new VA site rather than salvage; the other patient could not afford a stent graft.

2. Thrombosed and tortuous mega-fistula (n = 2, ang sub-protocol).

3. Ruptured ulcer of a pseudoaneurysm (n = 1, od sub-protocol).

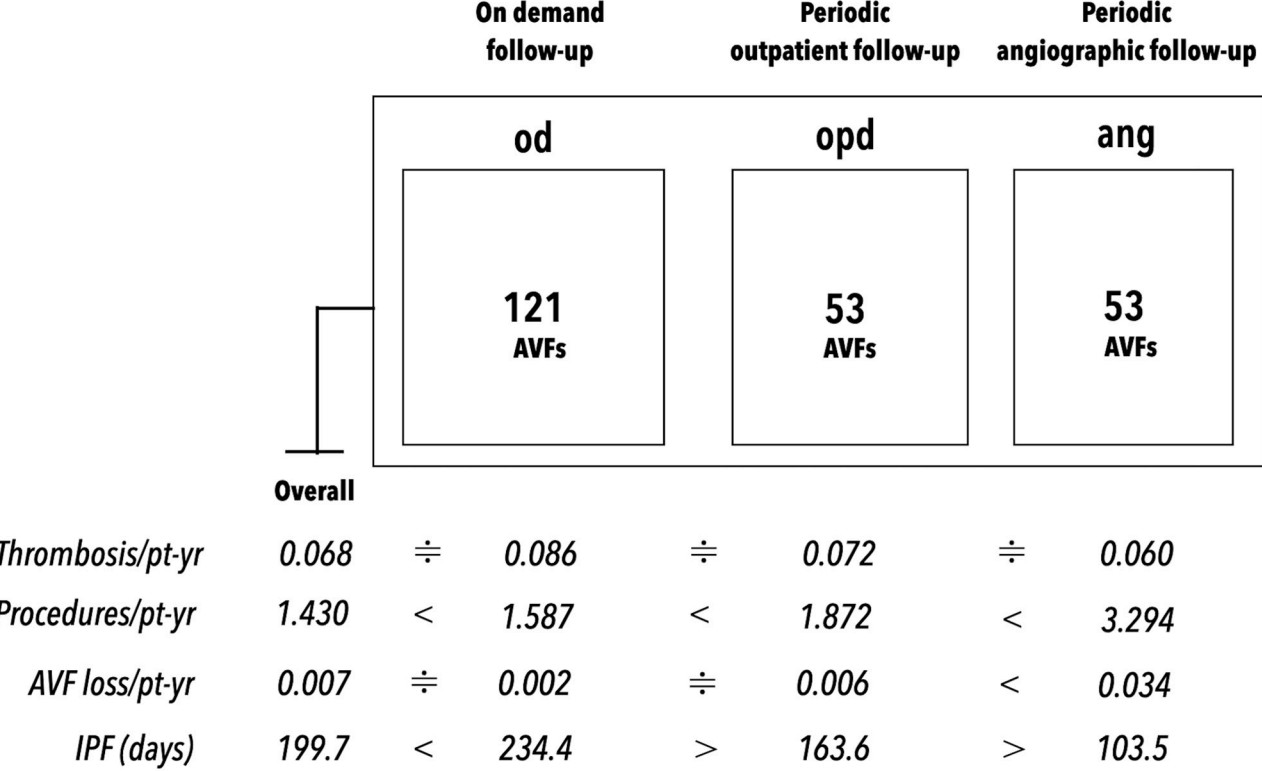

**Fig 4. Follow-up outcome measures of the protocol and sub-protocols.** Mathematical symbols: A>B or A<B imply that the value of A is statistically different from B; A÷B implies that the values of A and B were not statistically different. The thrombosis rate among the three sub-protocols was not statistically different, indicating the performances of the three sub-protocols for prevention of AVF thrombosis were equivalent.

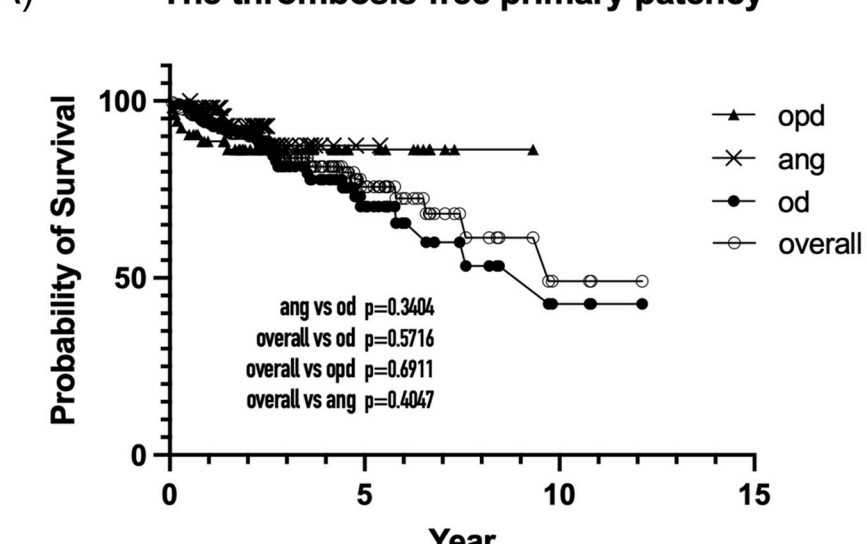

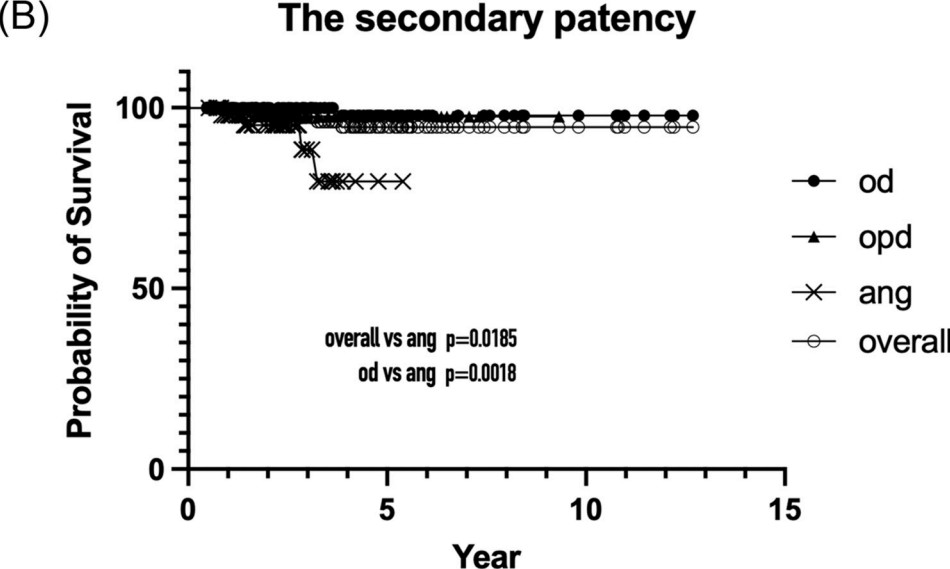

**Fig 5. The Kaplan–Meier survival curves for the follow-up protocol and sub-protocols.** (A) The thrombosis-free primary patency. No statistical differences were observed among the three follow-up sub-protocols. However, a declining trend was noticed for the curve of the od sub-protocol. (B) The secondary patency of the ang sub-protocol was statistically inferior to that of the od sub-protocol.

4. A thrombotic AVF with the eighth note deformity, where there was no recanalizable out-flow vein (n = 1, opd sub-protocol).

## Discussion

The primary findings of this study were as follows: 1) The protocol yielded an overall thrombosis rate of 0.068/pt-yr with a procedure rate of 1.43/pt-yr. There was no difference in the

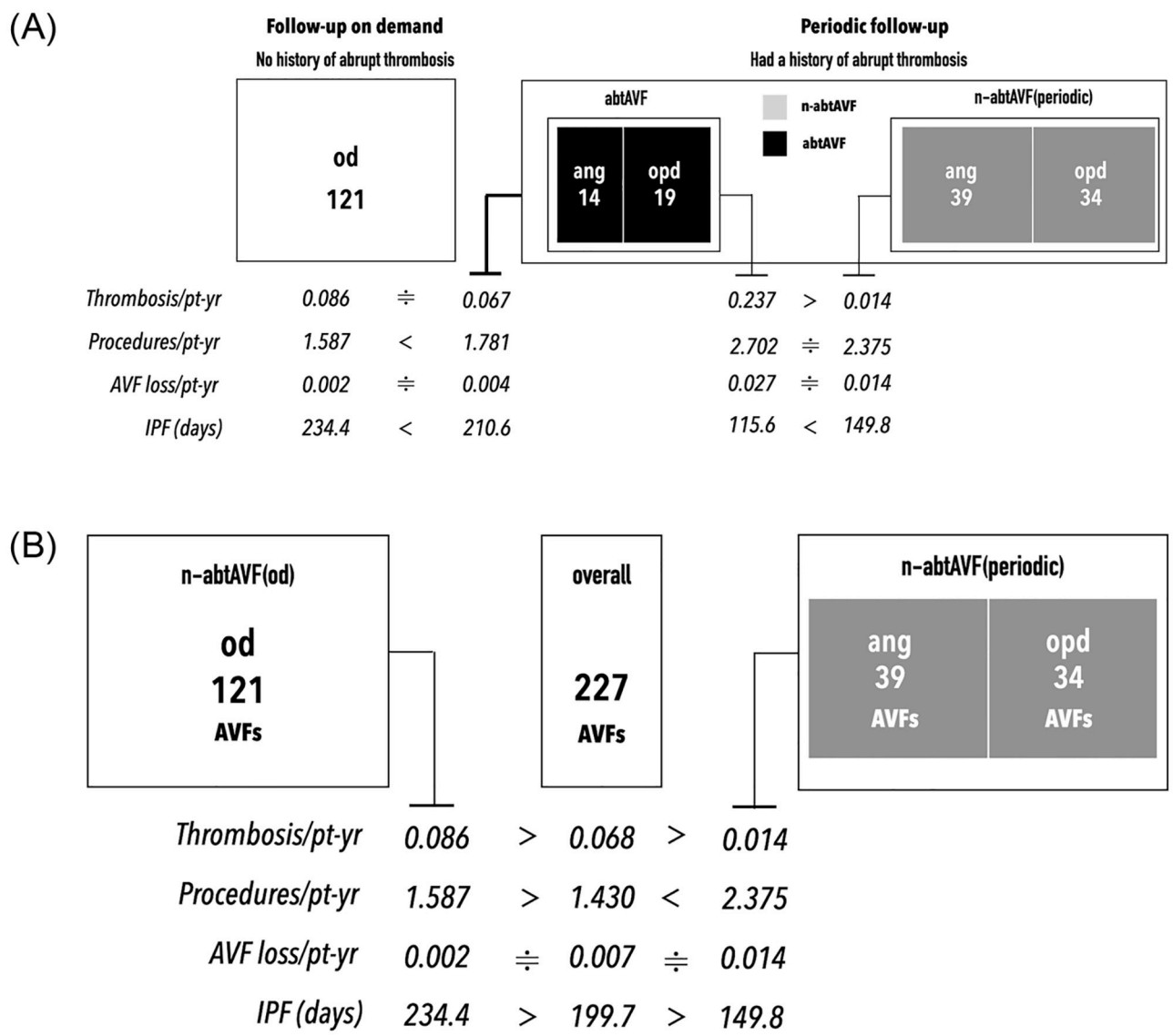

**Fig 6. Comparison of the outcome measures between target follow-up groups.** (A) Outcome measures of the abtAVF group and other target follow-up groups. Mathematical symbols: A>B or A<B imply that the value of A is statistically different from B; A≑B implies that the values of A and B were not statistically different. The thrombosis rate was highest in the abtAVF group and lowest in the n-abtAVF(periodic) group. There was no statistical difference in procedure rates between the two groups, indicating that the abtAVF group had a relatively faster restenosis rate than the n-abtAVF (periodic) group. (B) Comparison of the outcome measures between n-abtAVF(od)) vs. n-abtAVF(periodic). Mathematical symbols: A>B or A<B imply that the value of A is statistically different from B; A≑B implies that the values of A and B were not statistically different. For AVFs without a history of abrupt thrombosis, under periodic surveillance yielded a significantly lower thrombosis rate than that under on-demand follow-up.

thrombosis rate among the three sub-protocols. 2) The target rate of abtAVF was comparable with the protocol rate of the ang sub-protocol group. 3) The abtAVF group had significantly higher thrombosis and AVF loss rates, and a lower thrombosis-free primary patency compared with the n-abtAVF group. However, there were no significant differences in the secondary patency. 4) The n-abtAVF(periodic) group had the lowest thrombosis rate. 5) The n-abtAVF (od) group had a higher thrombosis rate, but a lower procedure rate compared with the n-abtAVF(periodic) group.

**Table 3. The thrombosis rate and thrombosis-free primary patency of the target follow-up groups.**

|  | Thrombosis rate/pt-yr | Thrombosis-free primary patency at 1, 4, and 8 years |
|---|---|---|
| **abtAVF** | 0.237 | 78.3%, 69.2%, 69.2% |
| **overall** | 0.068 | 94.1%, 80.8%, 60.9% |
| **n-abtAVF** | 0.042 | 96.8%, 82.9%, 62.3% |
| **n-abtAVF(od)** | 0.086 | 94.8%, 77.8%, 53.4% |
| **n-abtAVF(periodic)** | 0.014 | 100%, 98.3%, 98.3% |

To avoid AVF thrombosis, a follow-up protocol with a protocol rate greater than or equal to the target rate of the follow-up group should be selected to prevent AVF thrombosis. In the present study, the target rate of the abtAVF group and the protocol rate of the ang sub-protocol were similar, while the thrombosis rate of the abtAVF group was significantly higher than that of the ang sub-protocol. We believe this may have been attributed to a lower procedure rate in the abtAVF group. Among the 33 abtAVFs, only 14 were followed under the ang sub-protocol, while the others were followed under the opd sub-protocol. Under the opd sub-protocol, the interval from stenosis detection to angioplasty was approximately 2 weeks. Such a deferred correction of stenosis may lead to AVF thrombosis. Therefore, we believe that if the abtAVFs were followed under the ang sub-protocol, at the expense of an increased procedure rate, their thrombosis rates could have been reduced. That is, the appropriate IPF for the abtAVFs might be 103.5 days rather than 115.6 days.

Regarding the n-abtAVF group, if the goal of follow-up was to pursue a low thrombosis rate, the periodic sub-protocols (opd or ang) were considered appropriate because they yielded the lowest thrombosis rate. If patients wanted to undergo a procedure only when there was AVF dysfunction and the interventionist considered that their AVFs were not difficult to salvage once thrombosis occurred, the od sub-protocol was a reasonable option, because the goal of follow-up is to maintain an acceptably higher thrombosis rate compared with other target follow-up groups by following a sub-protocol with a lower procedure rate. In this study, AVFs without a history of abrupt thrombosis and under periodic follow-up (n-abtAVF(periodic)) had a high thrombosis-free or assisted primary patency of 100% at 1 year and 98% at 4 and 8 years. Aragoncillo et al. [8] also reported a high assisted primary patency of 91% at 1 year (estimated from the survival plot) for AVFs under flow surveillance. Tessitore et al. [2] reported a lower assisted primary patency of 85% at 1 year and 75% at 4 years (estimated from the survival plot) when subclinical stenoses in AVFs were detected using flow surveillance and treated pre-emptively. However, 30% of the AVFs in their treatment group were failing AVFs (Qa≤350 or recirculation>5%), which might explain their relatively low assisted primary patency.

According to Quencer et al. [9], VA site thrombosis accounts for 65%–85% of access loss. Therefore, lowering the thrombosis rate would reduce the risk of VA site loss. However, in the present study, the thrombosis rate of the abtAVF group was significantly higher than that of the n-abtAVF group, although their AVF loss rates were similar. This may have been attributed to recent innovations in endovascular techniques, which allowed us to handle challenging situations during the salvage procedure. Such innovations include endovascular bypass techniques for AVFs without recanalizable outflow veins, and the venotomy and manual propulsion technique for removal of massive thrombi [10–13]. Therefore, some VA sites may have been abandoned in the past but are now salvageable.

An important question is whether pursuing a low AVF thrombosis rate should be considered a patient-important outcome measure for AVF monitoring and surveillance. According to dissenters, surveillance leads to an increased procedure rate without confidence that it

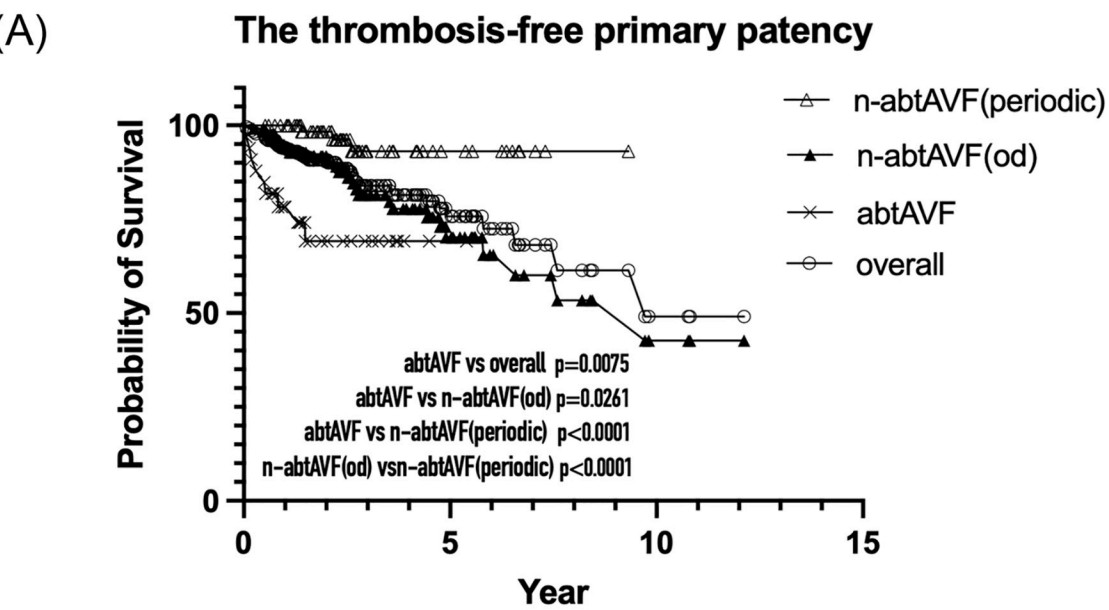

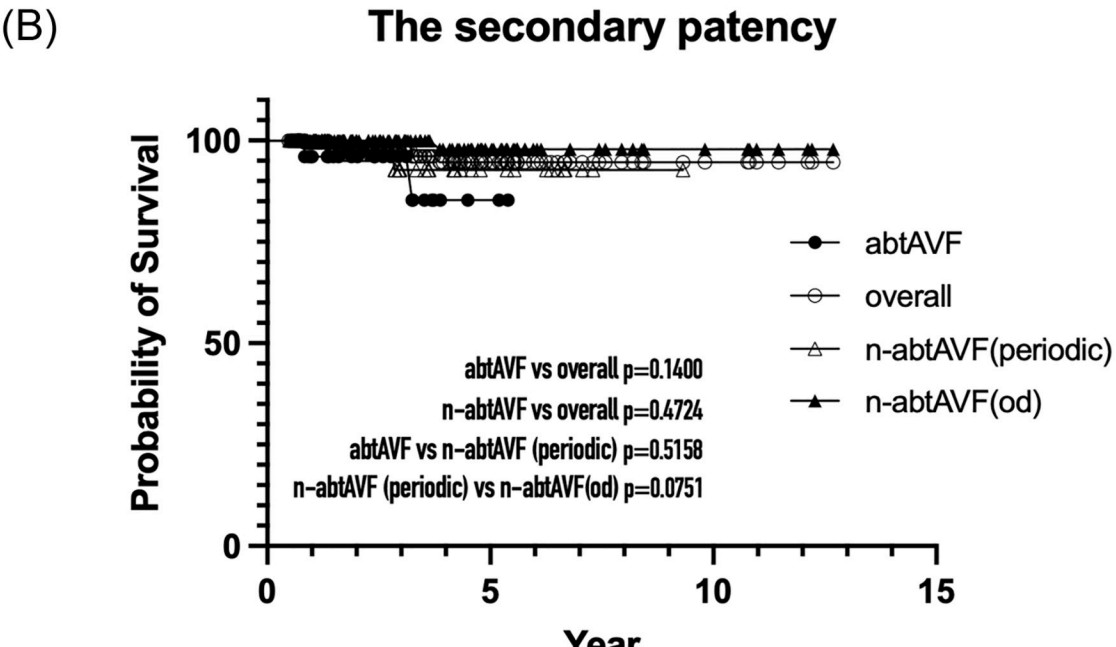

**Fig 7. The Kaplan–Meier survival curves for the target follow-up groups.** (A) The abtAVF group had the lowest 1-year thrombosis-free primary patency, and the n-abtAVF(periodic) group had the highest. (B) The secondary patency of the abtAVF group did not differ statistically from other target groups.

extends the life of the AVF as measured by secondary patency rates [14]. However, other studies reported that preemptive treatment of subclinical stenosis reduced the AVF loss rate and prolonged the functional life of AVFs [2].

From the perspectives of the patient and the interventionist, we believe that pursuing a low thrombosis rate should be considered a patient-important outcome measure. In addition to

**Table 4. The AVF loss rate and secondary patency of the target follow-up groups.**

|  | AVF loss rate/pt-yr | Secondary patency at 1 and 8 years |
|---|---|---|
| **abtAVF** | 0.027 | 96.0%, 85.3% |
| **overall** | 0.007 | 99.5%, 94.6% |
| **n-abtAVF** | 0.006 | 100%, 95.9% |
| **n-abtAVF(od)** | 0.002 | 100%, 97.8% |
| **n-abtAVF(periodic)** | 0.004 | 100%, 93.0% |

the pain at balloon dilatation and thrombus propulsion during the salvage procedure, there may be severe complications, such as venotomy wound infection and bleeding, hypovolemic shock, and pulmonary embolism. Additionally, compared with simply dilating stenoses, interventionists generally require more time and use more tools to salvage thrombotic VA sites. To salvage engorged and tortuous AVFs, especially those with more than one pseudoaneurysm, the procedure may take 2–3 h, and the risk of salvage failure is relatively high.

It may be controversial to recommend pursuing a low thrombosis rate for every VA site. We believe that pursuing a low thrombosis rate should be mandatory for the following populations of VA sites:

1. Salvage-challenging AVFs: These AVFs have the following features: a) a thrombotic, enlarged, and tortuous AVF with one or multiple pseudoaneurysms; b) a thrombotic AVF with heavy and extensive calcifications, especially around the perianastomotic area; and c) a thrombotic AVF without recanalizable outflow veins. Once a salvage-challenging AVF develops thrombosis, the odds of achieving successful salvage become very low. That is, for salvage-challenging AVFs, avoiding thrombosis prolongs their usable lives.

2. Patients who have nearly exhausted all available VA sites for hemodialysis.

In the present study, the overall thrombosis-free primary patency at 1 year was 94.1%. This was comparable with the results of AVFs followed by clinical monitoring combined with Qa surveillance (87%–97%) [2, 15, 16]. The 1-year thrombosis-free primary patency was lowest in the abtAVF group (78.3%). However, this was superior to results of AVFs followed up exclusively by clinical monitoring (61%–62%) [2, 15]. The AVF loss rate of the abrupt thrombosis group (0.027/pt-yr) was comparable with rates reported previously for AVFs followed by clinical monitoring combined with the Qa surveillance (0.024–0.066/pt-yr) [2, 17].

The present study had the following limitations. 1) The study was designed and conducted retrospectively. 2) There may have been recall bias in identifying abtAVFs. 3) The follow-up protocol and sub-protocols were not verified for follow-up of AV grafts. 4) The efficacy of the ang sub-protocol for follow-up of the abtAVF group requires verification with further study.

## Conclusions

We presumed that it might be appropriate to follow-up the AVFs with a history of abrupt thrombosis under the ang sub-protocol. For AVFs without a history of abrupt thrombosis, if the goal of follow-up is to pursue a low thrombosis rate, such as for salvage-challenging AVFs, they should be followed up periodically under the opd or ang sub-protocol. If the thrombotic AVFs were not difficult to salvage and patients want to undergo a procedure only when there is a positive clinical indicator, they can be followed up under the od sub-protocol.

## Author Contributions

**Conceptualization:** Matt Chiung-Yu Chen.

**Data curation:** Mei-Jui Weng.

**Formal analysis:** Mei-Jui Weng.

**Methodology:** Matt Chiung-Yu Chen, Lee-Hua Chao.

**Resources:** Lee-Hua Chao.

**Software:** Mei-Jui Weng.

**Supervision:** Matt Chiung-Yu Chen.

**Validation:** Matt Chiung-Yu Chen.

**Writing – original draft:** Matt Chiung-Yu Chen.

**Writing – review & editing:** Matt Chiung-Yu Chen.

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
