## [Decision Letter · Decision Letter 0]

30 Jan 2023

PONE-D-23-00790Characterization of Hemodialysis Fistulas Experienced Abrupt Thrombosis and Determination of a Proper Follow-up Protocol: A Retrospective Cohort Study and an Interventionist’s PerspectivePLOS ONE

Dear Dr. Chen,

Thank you for submitting your manuscript to PLOS ONE. After careful consideration, we feel that it has merit but does not fully meet PLOS ONE’s publication criteria as it currently stands. Therefore, we invite you to submit a revised version of the manuscript that addresses the points raised during the review process.

We look forward to receiving your revised manuscript.

Kind regards,

Redoy Ranjan, MBBS, MRCSEd, Ch.M., MS (CV&TS), FACS

Academic Editor

PLOS ONE

Journal Requirements:

Reviewers' comments:

Reviewer's Responses to Questions

**Comments to the Author**

1. Is the manuscript technically sound, and do the data support the conclusions?

Reviewer #1: Yes

Reviewer #2: Partly

Reviewer #3: Partly

2. Has the statistical analysis been performed appropriately and rigorously? 

Reviewer #1: Yes

Reviewer #2: N/A

Reviewer #3: I Don't Know

3. Have the authors made all data underlying the findings in their manuscript fully available?

Reviewer #1: Yes

Reviewer #2: Yes

Reviewer #3: No

4. Is the manuscript presented in an intelligible fashion and written in standard English?

Reviewer #1: Yes

Reviewer #2: Yes

Reviewer #3: No

5. Review Comments to the Author

Reviewer #1: Very nicely organized paper with well defenition for the subgroups. The follwup is very good and suffecient for all groups

No bias could be detedcted in the statistical nalaysis .

Overall very informative reserch paper.

Reviewer #2: The concept that artAVF has a lower patent rate can make sense, the reviewer has several comments:

1, The title is retrospective, but the method is prospective?

2, AVF and AVG are different, from the pathology viewpoint, the acute thrombosis has some difference, this data should be separated.

3, In the paper, the patent rat is 100% at the first year, and 98% at the 4th and 8th year, this is extremely higher than the commonly reported patent rate; and 8 years is a long time for CKD patient.

4, Are these patients received anticoagulation therapy? Should be present the data and discuss.

5, How could the authors discriminate the technique failures induced thrombosis?

6, What is the diameters of the artery and vein?

7, What is the diameter of the prosthetic graft? and what is the materials?

Reviewer #3: The present manuscript has potentially important data on an interesting topic how to get better patency for failing artery - venous fistula. The cohort is reasonable sized 227 patients. The main issue with the manuscript is that it would have been easier to review if authors had finalized the manuscript little more. The order of presented issues could be more logical, results and observations should be more accurately presented, tables should have all appendixes explained and legends be more detailed and even figure legends were not on a file for the review.

Also it would be interesting to have a demographic data correlated with the AVF requiring additional procedures and the ones that function just fine.

Abstract: Some parts of the abstract are hard to follow and AVF both with and without thrombosis should be clearly defined when presenting results.

Introduction: There should be a strong body of literature on AVF thrombosis and patency, relevant and as scientific base for the introduction. At present introduction lacks the scientific touch.

Methods: Sample characteristics with the statement of ethical approval details etc. would define the work better? The methods could be better formulated to be easier to read and interpret. Also, some of the nice flow charts could be in supplement? Essentially at present methods is hard to follow and still does not include required details.

Results: Over 500 VA and only little over 227 enrolled rationales? Figure 3 shows the explanation, but authors should either refer to figure 3 after stating 227 enrolled or explain briefly why half of the VA were excluded. It would be nice to have more detailed demographic data presented.

Tables should have subheading containing the appendixes written out and be more comprehensive.

Figures should have detailed legend? Now these legends seem to be missing.

The discussion should be more fluent.

6. PLOS authors have the option to publish the peer review history of their article (what does this mean?). If published, this will include your full peer review and any attached files.

Reviewer #1: **Yes: **Aram Baram

Reviewer #2: No

Reviewer #3: No

---

## [Author Response · Author response to Decision Letter 0]

5 Feb 2023

Author’s response to reviews

Title: Characterization of hemodialysis fistulas experienced abrupt thrombosis and determination of a proper follow-up protocol: A retrospective cohort study and an interventionist’s perspective

Authors:

Matt Chiung-Yu Chen (jjychen@gmail.com)

Mei-Jui Weng (mjweng@gmail.com)

Lee-Hua Chao (y0053@yuanhosp.com.tw)

Version: 1, Date 5th Feb 2023

Author’s response to reviews

Manuscript number: PONE-S-23-01053. R1

Journal Requirements:

1. Please ensure that your manuscript meets PLOS ONE's style requirements, including those for file naming

A: The manuscript has been thoroughly reviewed to ensure it meets PLOS ONE's style requirements.

A: A paragraph addressing the data availability of this study has been added to the revised cover letter as follows: “The authors confirm that all data underlying the findings are fully available without restriction. All data have been deposited to Figshare (doi: 10.6084/m9.figshare.21996920).”

A: A sub-section entitled "Patient population and study design" has been added to the Methods section as follows: “Between November 2020 and February 2021, patients referred to our institution for treatment of vascular access (VA) sites were enrolled in this study. All VA sites were treated and followed according to our routine protocols. After obtaining approval from our hospital's institutional review board committee (IRB number: 20220011B), the patients' electronic imaging and medical records were reviewed. Data for this study was collected prospectively and routinely recorded after each treatment was completed. To conduct this study, the collected data was retrieved and analyzed retrospectively.”

Reviewer #2:

1. The title is retrospective, but the method is prospective?

A: The authors are sorry for the ambiguity. The data in this study was collected prospectively, which was routinely recorded after each treatment was completed. To conduct this study, the collected data was retrieved analyzed retrospectively.

2. AVF and AVG are different, from the pathology viewpoint, the acute thrombosis has some difference, this data should be separated. What is the diameter of the prosthetic graft? and what is the materials? +

A: We are sorry for the ambiguity. The study, as stated in the Methods section and shown in Fig 3, only autogenous AVFs were included. All AVGs were excluded per the exclusion criteria of this study.

3. In the paper, the assisted primary patency rate for n-abtAVF(periodic) is 100% at the first year, and 98% at the 4th and 8th year, this is extremely higher than the commonly reported patent rate; and 8 years is a long time for CKD patient.

A: Thanks for your pertinent comments. A paragraph addressing this issue had been added into the Discussion section as follows:” In this study, AVFs without a history of abrupt thrombosis and under periodic follow-up (n-abtAVF(periodic)) had a high thrombosis-free or assisted primary patency of 100% at 1 year and 98% at 4 and 8 years. Aragoncillo et al. also reported a high assisted primary patency of 91% at 1 year (estimated from the survival plot) for AVFs under flow surveillance. Tessitore et al reported a lower assisted primary patency of 85% at 1 year and 75% at 4 years (estimated from the survival plot) when subclinical stenoses in AVFs were detected using flow surveillance and treated preemptively. However, 30% of the AVFs in their treatment group were failing AVFs (Qa≤350 or recirculation>5%), which might explain their relatively low assisted primary patency.”

4. Are these patients received anticoagulation therapy? Should be present the data and discuss.

A: Thanks for your comments. The duration and types of anticoagulants vary for the patients in this study. The question is very difficult to answer. For example, if a patient takes anticoagulants only for a few days or weeks during a three-year follow-up period, yes, he received anticoagulation therapy but it is nearly impossible to assess the effect of such an anti-coagulation regimen on his AVF patency. Moreover, our aim is to discuss about AVF, and discussing the effect of anticoagulants on AVF patency would make the Discussion section out of focus.

5. How could the authors discriminate the technique failures induced thrombosis?

A: Thanks for your comments. We didn’t mention technique failures induced thrombosis in this manuscript but technical failure for salvage of a thrombotic AVF, especially the salvage -challenging AVFs.

6. What are the diameters of the artery and vein?

A: Thanks for your comments. For a given AVF, the diameter varies from the anastomosis to the outflow veins. Could you specify your question?

Reviewer #3:

1. The order of presented issues could be more logical, results and observations should be more accurately presented.

A: Thanks for your comments. The Methods section has been extensively rewritten and the order of its sub-sections have been re-arranged.

2. Tables should have all appendixes explained and legends be more detailed and even figure legends were not on a file for the review. Tables should have subheading containing the appendixes written out and be more comprehensive. Figures should have detailed legend? Now these legends seem to be missing.

A: Thanks for your pertinent comments. The appendixes of Table 2 have been written out. Detailed figure legends were added to Figs 4~7.

3. Also it would be interesting to have a demographic data correlated with the AVF requiring additional procedures and the ones that function just fine.

A: The authors agree with you that it would be interesting to correlate demographic data with AVFs that require additional procedures and those that functioning well without intervention. However, we do not have approval from the Institutional Review Board (IRB) to review those well-functioning AVFs that do not require any intervention.

4. Abstract: Some parts of the abstract are hard to follow and AVF both with and without thrombosis should be clearly defined when presenting results.

A: Due to the word count limit of 300-500 words for the Abstract section, we cannot provide too much detail. Definitions have been included in the Definitions subsection of the Methods section.

5. Introduction: There should be a strong body of literature on AVF thrombosis and patency, relevant and as scientific base for the introduction. At present introduction lacks the scientific touch.

A: Thank you for your insightful comments. The Introduction section has been thoroughly rewritten and the necessary scientific background has been provided.

Three new paragraphs have been added: the first one addressed the "dysfunctional hypothesis", the second discussed the presence of subclinical stenosis, and the third focused on the thrombosis-high-risk stenosis and its implications for AVF abrupt thrombosis.

6. Methods: Sample characteristics with the statement of ethical approval details etc. would define the work better? The methods could be better formulated to be easier to read and interpret.

A: Thanks for your comments. We have revised the Methods section to make it easier to read and interpret.

7. Also, some of the nice flow charts could be in supplement? Essentially at present methods is hard to follow and still does not include required details.

A: Thanks for your suggestion. The authors prefer to keep Figures 1-3 in their current locations rather than moving them to the Supporting Information section. This would be convenient for readers, as they may want to view the detailed sub-protocol flow charts after reading the sub-protocol definitions, and they can access these flow charts easily.

8. Results: Over 500 VA and only little over 227 enrolled rationales? Figure 3 shows the explanation, but authors should either refer to figure 3 after stating 227 enrolled or explain briefly why half of the VA were excluded.

A: Thanks for your comments. The paragraph has been rewritten as follows: “A total of 505 VA sites in 505 patients were treated during the study period. Of them, 227 AVFs were enrolled. The study flow diagram is shown in Figure 3. Most of the excluded VA sites were AVGs, n=149 (29.5%).”

9. The discussion should be more fluent.

A: Thanks for your comments. We have added a paragraph to the Discussion section addressing the high thrombosis-free primary patency of the n-abtAVF(periodic) group, which should make the section easier to follow. Additionally, a professional editor has edited the manuscript.http://www.intervrad.net/shareCenter/proof.jpg

---

## [Decision Letter · Decision Letter 1]

27 Feb 2023

Characterization of Hemodialysis Fistulas Experienced Abrupt Thrombosis and Determination of a Proper Follow-up Protocol: A Retrospective Cohort Study and an Interventionist’s Perspective

PONE-D-23-00790R1

Dear Dr. Chen,

We’re pleased to inform you that your manuscript has been judged scientifically suitable for publication and will be formally accepted for publication once it meets all outstanding technical requirements.

Kind regards,

Redoy Ranjan, MBBS, MRCSEd, Ch.M., MS (CV&TS), FACS

Academic Editor

PLOS ONE

Additional Editor Comments (optional):

Review Comments to the Author

Reviewer #1: (No Response)

Reviewer #2: The authors have addressed all of the reviewer's concerns. the quality of the paper is now improved,

Reviewer #3: All given comments have been addressed. The present manuscript has been improved significantly after revision

---

## [Editor Report · Acceptance letter]

2 Mar 2023

PONE-D-23-00790R1 

Characterization of hemodialysis fistulas experienced abrupt thrombosis and determination of a proper follow-up protocol: A retrospective cohort study and an interventionist’s perspective 

Dear Dr. Chen:

I'm pleased to inform you that your manuscript has been deemed suitable for publication in PLOS ONE. Congratulations! Your manuscript is now with our production department. 

Kind regards, 

on behalf of

Dr. Redoy Ranjan 

Academic Editor

PLOS ONE